# The Multiscale Surface Vision Transformer

**Simon Dahan**[1]                                                          SIMON.DAHAN@KCL.AC.UK
**Logan Z. J. Williams**[1,2]                                              LOGAN.WILLIAMS@KCL.AC.UK
**Daniel Rueckert**[3]                                                     DANIEL.RUECKERT@TUM.DE
**Emma C. Robinson**[1,2]                                                  EMMA.ROBINSON@KCL.AC.UK

[1] *Department of Biomedical Engineering & Imaging Science, King's College London*

[2] *Centre for the Developing Brain, King's College London*

[3] *Institute for AI in Medicine, Technical University of Munich*

**Editors:** Accepted for publication at MIDL 2024

## Abstract

Surface meshes are a favoured domain for representing structural and functional information on the human cortex, but their complex topology and geometry pose significant challenges for deep learning analysis. While Transformers have excelled as domain-agnostic architectures for sequence-to-sequence learning, the quadratic cost of the self-attention operation remains an obstacle for many dense prediction tasks. Inspired by some of the latest advances in hierarchical modelling with vision transformers, we introduce the Multiscale Surface Vision Transformer (MS-SiT) as a backbone architecture for surface deep learning. The self-attention mechanism is applied within local-mesh-windows to allow for high-resolution sampling of the underlying data, while a shifted-window strategy improves the sharing of information between windows. Neighbouring patches are successively merged, allowing the MS-SiT to learn hierarchical representations suitable for any prediction task. Results demonstrate that the MS-SiT outperforms existing surface deep learning methods for neonatal phenotyping prediction tasks using the Developing Human Connectome Project (dHCP) dataset. Furthermore, building the MS-SiT backbone into a U-shaped architecture for surface segmentation demonstrates competitive results on cortical parcellation using the UK Biobank (UKB) and manually-annotated MindBoggle datasets. Code and trained models are publicly available at https://github.com/metrics-lab/surface-vision-transformers.

**Keywords:** Vision Transformers, Cortical Imaging, Geometric Deep Learning, Segmentation, Neurodevelopment

## 1. Introduction

In recent years, there has been an increasing interest in using attention-based learning methodologies in the medical imaging community, with the Vision Transformer (ViT) (Dosovitskiy et al., 2020) emerging as a particularly promising alternative to convolutional methods. The ViT circumvents the need for convolutions by translating image analysis to a sequence-to-sequence learning problem, using self-attention mechanisms to improve the modelling of long-range dependencies. This has led to significant improvements in many medical imaging tasks, where global context is crucial, such as tumour or multi-organ segmentation (Tang et al., 2021; Ji et al., 2021; Hatamizadeh et al., 2022). At the same time, there has been a growing enthusiasm for adapting attention-based mechanisms to irregular geometries where the translation of the convolution operation is not trivial, but the

representation of the data as sequences can be straightforward, for instance for protein modelling (Atz et al., 2021; Jumper et al., 2021; Baek et al., 2021) or functional connectomes (Kim et al., 2021). Similarly, vision transformers (ViTs) have been recently translated to the study of cortical surfaces (Dahan et al., 2022), by re-framing the problem of surface analysis on sphericalised meshes as a sequence-to-sequence learning task and by doing so improving the modelling of long-range dependencies in cortical surfaces. Transformer models have also emerged as a promising tool for modelling various cognitive processes, such as language and speech (Millet et al., 2022; Défossez et al., 2023), vision (Tang et al., 2023), and spatial encoding in the hippocampus (Whittington et al., 2021).

Despite promising results on high-level prediction tasks, one of the main limitations of the ViT remains the computational cost of the global self-attention operation, which scales quadratically with sequence length. This limits the ability of the ViT to capture fine-grained details and to be used directly for dense prediction tasks. Various strategies have been developed to overcome this limitation, including restricting the computation of self-attention to local windows (Fan et al., 2021; Liu et al., 2021) or implementing linear approximations (Wang et al., 2020; Xiong et al., 2021). Among these, the hierarchical architecture of the Swin Transformer (Liu et al., 2021) has emerged as a particularly favoured candidate. This implements windowed local self-attention, alongside a shifted window strategy that allows cross-window connections. Neighbouring patch tokens are progressively merged across the network, producing a hierarchical representation of image features. This hierarchical strategy has shown to improve performance over the global-attention approach of the standard ViT, and has already found applications within the medical imaging domain (Hatamizadeh et al., 2022). Cheng et al. (2022) attempted to adapt such windowed local attention to the study of cortical meshes. However, attention windows were defined as the vertices forming the hexagonal patches of a low-resolution grid, but not the patch features. This restricts the feature extraction with self-attention to a small number of vertices on the mesh and greatly limits the local feature extraction capabilities of the model.

In this paper, we therefore introduce the Multiscale Surface Vision Transformer (MS-SiT) as a novel backbone architecture for surface deep learning. The MS-SiT takes inspiration from the Swin Transformers model and extends the Surface Vision Transformers (SiT) (Dahan et al., 2022) to a hierarchical version that can serve for any high-level or dense prediction task on sphericalised meshes. First, the MS-SiT introduces a local-attention operation between surface patches and within local attention-windows defined by the subdivisions of a high-resolution sampling grid. This allows for the modelling of fine-grained details of cortical features (with sequences of up to 20,480 patches). Moreover, to preserve the modelling of long-range dependencies between distant regions of the input surface, the MS-SiT adapts the shifted local-attention approach, introduced in (Liu et al., 2021), by shifting the sampling grid across the input surface. This allows propagation of information between neighbouring attention-windows, achieving global attention at a reduced computational cost; however, it is challenging to implement due to the irregular spacing and sampling of vertices on native surface meshes. We evaluate our approach on neonatal phenotype prediction tasks derived from the Developing Human Connectome Project (dHCP), as well as on cortical parcellation for both UK Biobank (UKB) and manually-annotated MindBoggle datasets. Our proposed MS-SiT architecture strongly surpasses existing surface deep learning methods for predictions of cortical phenotypes and achieves competitive performance

on cortical parcellation tasks, highlighting its potential as a holistic deep learning backbone and a powerful tool for clinical applications.

## 2. Methods

**Backbone** The proposed MS-SiT adapts the Swin Transformer architecture (Liu et al., 2021) to the case of cortical surface analysis, as illustrated in Figure 1. Here, input data $X \in \mathbb{R}^{|V_6| \times C}$ ($C$ channels) is represented on a 6th-order icospheric (ico6) tessellation: $I_6 = (V_6, F_6)$, with $|V_6| = 40962$ vertices and $|F_6| = 40962$ faces. This data is first partitioned into a sequence of $|F_5| = 20480$ non-overlapping triangular patches: $T_5 = \{t_5^1, t_5^2, ..t_5^{|F_5|}\}$ (with $t_5^i \subset V_6, |t_5^i| = |t_5| = 6$), by patching the data with ico5: $I_5 = (V_5, F_5), |V_5| = 10242, |F_5| = 20480$ (Figure 1 A.2). Imaging features for each patch are then concatenated across channels, and flattened to produce an initial sequence: $X^0 = \left[ X_1^0, ..., X_{|F_5|}^0 \right] \in \mathbb{R}^{|F_5| \times (C|t_5|)}$ (Figure 1A.3). Trainable positional embeddings, LayerNorm (LN) and a dropout layer are then applied, before passing it to the MS-SiT encoder, organised into $l = \{1, 2, 3, 4\}$ levels.

At each level of the encoder, a linear layer projects the input sequence $X^l$ to a $2^{(l-1)} \times D$-dimensional embedding space: $X_{emb}^l \in \mathbb{R}^{|F_{6-l}| \times 2^{(l-1)}D}$. Local multi-head self-attention blocks (local-MHSA), described in section 2, are then applied, outputting a transformed sequence of the same resolution ($X_{MHSA}^l \in \mathbb{R}^{|F_{6-l}| \times 2^{(l-1)}D}$). This is subsequently down-sampled through a patch merging layer, which follows the regular downsampling of the icosphere, to merge clusters of 4 neighbouring triangles together (Figure 1B), generating output: $X_{out}^l \in \mathbb{R}^{|F_{6-l-1}| \times 2^{(l+1)}D}$.

This process is repeated across several layers, with the spatial resolution of patches progressively downsampled from $I_5 \to I_4 \to I_3 \to I_2$, but the channel dimension doubling each time. In doing so, the MS-SiT architecture produces a hierarchical representation of patch features, with respectively $|F_5| = 20480$, $|F_4| = 5120$, $|F_3| = 1280$, and $|F_2| = 320$ patches. In the last level, the patch merging layer is omitted (see Figure 1) and the sequence of patches is averaged into a single token, and input to a final linear layer, for classification or regression (Figure 1A.5). Inspired by previous work (Cao et al., 2021), the segmentation pipeline employs a UNet-like architecture, with skip-connections between encoder and decoder layers, and patch partition instead of patch merging applied during decoding. An illustration of the pipeline is provided in Figure 3, Appendix A.

**Local Multi-Head Self-Attention blocks** are defined similarly to ViT blocks (Dosovitskiy et al., 2020): as successive multi-head self-attention (MHSA) and feed-forward (FFN) layers, with LayerNorm (LN) and residual layers in between (Figure 1C). Here, a **W**indow-MHSA (**W-MHSA**) replaces the global MHSA of standard vision transformers, applying self-attention between patches within non-overlapping local mesh-windows. To provide the model with sufficient contextual information, this attention window is defined by an icosahedral tessellation three levels down from the resolution used to represent the feature sequence. This means that at level $l$, while the sequence is represented by $I_{6-l}$, the attention windows correspond to the non-overlapping faces $F_{6-(l+3)}$ defined by $I_{6-(l+3)}$. For example, at level 1 the features are input at ico5, and local attention is calculated between the subset of 64 triangular patches that overlap with each face of ico2 ($F_2$), see Figure 1.B.1. Only in the last layer, is attention not restricted to local windows but applied globally to the $I_2$ grid,

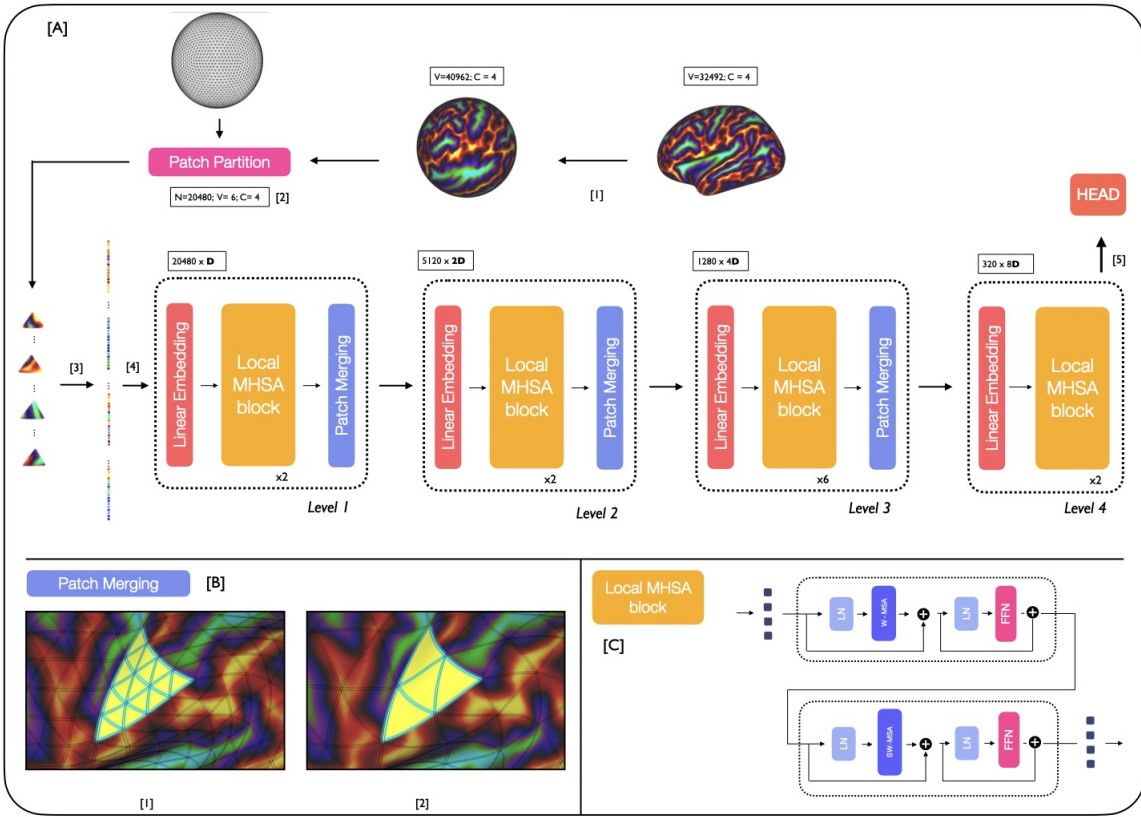

Figure 1: [A] MS-SiT pipeline. The input cortical surface is resampled from native resolution (1) to an ico6 input mesh and partitioned (2). The sequence is then flattened (3) and passed to the MS-SiT encoder layers (4). The head (5) can be adapted for classification or regression tasks. [B] illustrates the patch merging operation (here from $I_4$ to $I_3$ grid). High-resolution patches are grouped by 4 to form patches of lower-resolution sampling grid [C] A Local-MHSA block is composed of two attention blocks: **W**indow-MHSA and **S**hifted **W**indow-MHSA.

allowing for global sharing of information across the entire sequence. More details of the parameterisation of window attention grids is provided in the Appendix A, Table 3. This use of local self-attention significantly reduces the computational cost of attention at level $l$, from $\mathcal{O}(|F_{6-l}|^2)$ to $\mathcal{O}(w_l|F_{6-l}|)$ with $w_l << |F_{6-l}|$.

**Self-Attention with Shifted Windows** Cross-window connections are introduced through **S**hifted **W**indow MHSA (**SW-MHSA**) modules, to improve the modelling power of the local self-attention operations. These alternate with the W-MHSA, and are implemented by shifting all the patches in the sequence $I_{6-l}$, at level $l$ by $w_s$ positions, where $w_s$ is a fraction of the window size $w_l$ (typically $w_l = 64$). In this way, a fraction of the patches of each attention window now falls within an adjacent window (see Figure 4). This preserves

the cost of applying self-attention in a windowed fashion, whilst increasing the models representational power by sharing information between non-overlapping attention windows.

The W-MHSA and SW-MHSA implementation can be summarised as follows:

$$\begin{aligned}
\hat{\mathbf{X}}^l &= W\text{-}MSA(\mathbf{X}^l_{emb}) + \mathbf{X}^l_{emb} \\
\mathbf{Z}^l &= FFN(\hat{\mathbf{X}}^l) + \hat{\mathbf{X}}^l \\
\hat{\mathbf{Z}}^l &= SW\text{-}MSA(\mathbf{Z}^l) + \mathbf{Z}^l \\
\mathbf{X}^l_{MHSA} &= FFN(\hat{\mathbf{Z}}^l) + \hat{\mathbf{Z}}^l
\end{aligned} \tag{1}$$

Here $\mathbf{X}^l_{emb}$ and $\mathbf{X}^l_{MHSA}$ correspond to input and output sequences of the local-MHSA block at level $l$. Residual connections are referred to by the $+$ symbol.

**Training details** Augmentation strategies were introduced to improve regularisation and increase transformation invariance. This included implementing random rotational transforms, where the degree of rotation about each axis was randomly sampled in the range $\in [-30°, +30°]$ (for the regression tasks) and $\in [-15°, +15°]$ (for the segmentation tasks). In addition, elastic deformations were simulated by randomly displacing the vertices of a coarse ico2 grid to a maximum of 1/8th of the distance between neighbouring points (to enforce diffeomorphisms (Fawaz et al., 2021)). These deformations were interpolated to the high-resolution grid of the image domain, online, during training. The effect of tuning the parameters of the SW-MHSA modules is presented in Table 4 and reveals that the best results are obtained while shifting half of the patches.

## 3. Experiments & Results

All experiments were run on a single RTX 3090 24GB GPU. The AdamW optimiser (Loshchilov and Hutter, 2017) with Cosine Decay scheduler was used as the default optimisation scheme, more details about optimisation and hyper-parameters tuning in Appendix B.2. A combination of Dice Loss and CrossEntropyLoss was used for the segmentation tasks and MSE loss was used for the regression tasks. Surface data augmentation was randomly applied with a probability of 80%. If selected, one random transformation is applied: either rotation (50%) or non-linear warping (50%). For all regression tasks, a custom balancing sampling strategy was applied to address the imbalance of the data distribution.

### 3.1. Phenotyping predictions on dHCP data

**Data** from the dHCP comes from the publicly available third release[1] (Edwards et al., 2022) and consists of cortical surface meshes and metrics (sulcal depth, curvature, cortical thickness and T1w/T2w myelination) derived from T1- and T2-weighted magnetic resonance images (MRI), using the dHCP structural pipeline, described by (Makropoulos et al., 2018) and references therein (Kuklisova-Murgasova et al., 2012; Schuh et al., 2017; Hughes et al., 2017; Cordero-Grande et al., 2018; Makropoulos et al., 2018). In total 580 scans were used from 419 term neonates (born after 37 weeks gestation) and 111 preterm neonates (born prior to 37 weeks gestation). 95 preterm neonates were scanned twice, once shortly after birth, and once at term-equivalent age.

---

1. http://www.developingconnectome.org.

| Model | Aug. | Shifted Attention | PMA Template | PMA Native | GA Template | GA Native |
|-------|------|-------------------|--------------|------------|-------------|-----------|
| SUNet [2] | ✓ | n/a | 0.75±0.18 | 1.63±0.51 | 1.14±0.17 | 2.41±0.68 |
| MoNet [3] | ✓ | n/a | 0.61±0.04 | 0.63±0.05 | 1.50±0.08 | 1.68±0.06 |
| SiT-T (ico2) | ✓ | n/a | 0.58±0.02 | 0.66±0.01 | 1.04±0.04 | 1.28±0.06 |
| SiT-T (ico3) | ✓ | n/a | 0.54±0.05 | 0.68±0.01 | 1.03±0.06 | 1.27±0.05 |
| SiT-T (ico4) | ✓ | n/a | 0.57±0.03 | 0.83±0.04 | 1.41±0.09 | 1.49±0.10 |
| MS-SiT | ✓ | ✗ | **0.49±0.01** | **0.59±0.01** | 1.00±0.04 | 1.17±0.04 |
| MS-SiT | ✓ | ✓ | **0.49±0.01** | **0.59±0.01** | **0.88±0.02** | **0.93±0.05** |

Table 1: Test results for the dHCP phenotype prediction tasks: PMA and GA. **M**ean **A**bsolute **E**rror (MAE) and std are averaged across three training runs for all experiments.

**Tasks and experimental set up:** Phenotype regression was benchmarked on two tasks: prediction of postmenstrual age (PMA) at scan, and gestational age (GA) at birth. Here, PMA was seen as a model of 'healthy' neurodevelopment, since training data was drawn from the scans of term-born neonates and preterm neonates' first scans: covering brain ages from 26.71 to 44.71 weeks PMA. By contrast, the objective of the GA model was to predict the degree of prematurity (birth age) from the participants' term-age scans, thus the model was trained on scans from term neonates and preterm neonates' second scans. Experiments were run on both registered (***template*** space) and unregistered (***native*** space) data to evaluate the generalisability of MS-SiT compared to surface convolutional approaches (Spherical UNet (SUNet) (Zhao et al., 2019) and MoNet (Monti et al., 2016)). The four aforementioned cortical metrics were used as input data. Training test and validation sets were allocated in the ratio of 423:53:54 examples (for PMA) and 411:51:52 (for GA) with a balanced distribution of examples from each age bin.

**Results** from the phenotyping prediction experiments are presented in Table 1, where the MS-SiT models were compared against several surface convolutional approaches and three versions of the Surface Vision Transformer (SiT) using different grid sampling resolutions. The MS-SiT model consistently outperformed all other models across all prediction tasks (PMA and GA) and data configurations (template and native). Specifically, on the PMA task, the MS-SiT model outperformed other models by over 54% compared to Spherical UNet (Zhao et al., 2019), 13% to MoNet (Monti et al., 2016), and 12% to the SiT (ico3) (average over both data-configurations), achieving a prediction error of 0.49 MAE on template data, which is within the margin of error of age estimation in routine ultrasound (typically, 5 to 7 days on average). On the GA task, the MS-SiT model achieved an even larger improvement with 49%, 43%, and 21% reduction in MAE relative to Spherical UNet, MoNet, and SiT (ico3), respectively. Importantly, the model demonstrated much greater transformation invariance, with only a 5% drop in performance between the template and native configurations, compared to 53% for Spherical UNet, and 10% for MoNet. Results

---

2. (Zhao et al., 2019)
3. (Monti et al., 2016)

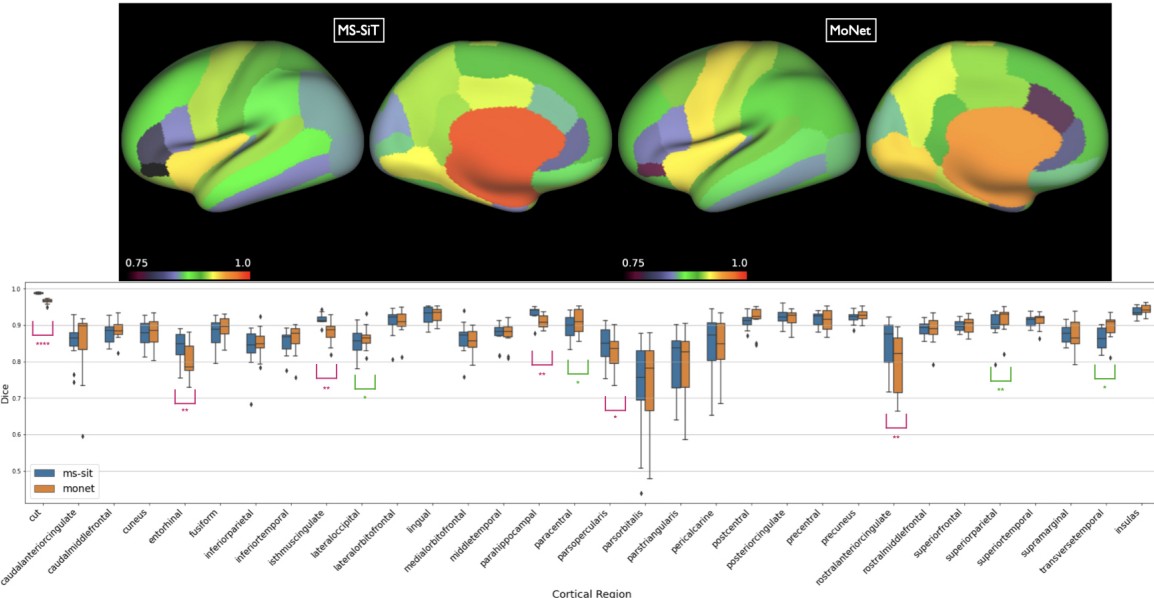

Figure 2: Top: Inflated surface showing mean dice scores shown for each of the DKT regions, for both MoNet and the pre-trained MS-SIT. Bottom: Boxplots comparing regional parcellation results. Asterisks denote statistical significance for one-sided paired t-test (pink: MS-SiT > MoNet; green MoNet > MS-SIT; ****: $p < 0.0001$, **: $p < 0.01$, *: $p < 0.05$ ).

also revealed a significant benefit to using the SW-MHSA with a 16% improvement over the vanilla version on GA predictions.

### 3.2. Cortical parcellation on UKB & MindBoggle

**Data & Tasks** Cortical segmentation was performed using 88 manually labelled adult brains from the MindBoggle-101 dataset (Klein and Tourville, 2012)[4], annotated into 31 regions using a modified version of the Desikan–Killiany (DK) atlas (Desikan et al., 2006), which delineates regions according to features of cortical shape. Surface files were processed with the Ciftify pipeline (Dickie et al., 2019), which implements HCP-style post-processing including file conversion to GIFTI and CIFTI formats, and MSM Sulc alignment (Robinson et al., 2014, 2018) [5]. Separately, FreeSurfer annotation parcellations (based on a standard version of the DK atlas with 35 regions) were available for 4000 UK Biobank subjects, processed according to (Alfaro-Almagro et al., 2018). These were used for pretraining. In both cases, datasets were split into 80%/10%/10% sets. As the annotations characterise folding patters, we used shape-based cortical metrics as input features: sulcal depth and curvature maps.

---

4. https://MindBoggle.info/data
5. 13 datasets failed due to missing files

| Methods | Augmentation | Dice overlap |
|---|---|---|
| Adv-GCN[6] | n/a | 0.857±0.04 |
| SPHARM-Net [7] | n/a | 0.887±0.06 |
| MoNet [8] | ✓ | **0.910±0.01** |
| MS-SiT | ✓ | 0.897±0.01 |
| MS-SiT (UKB) | ✓ | 0.901±0.01 |

Table 2: Overall mean and standard deviation of Dice scores (across all regions).

**Results** are presented in Table 2. The MS-SiT was compared against three other gDL approaches for cortical segmentation: Adv-GCN, a graph-based method optimized for alignment invariance (Gopinath et al., 2020), SPHARM-net (Ha and Lyu, 2022), a spherical harmonic-based CNN method, and MoNet, which learns filters by fitting mixtures of Gaussians on the surface (Monti et al., 2016). MoNet achieved the best dice results overall, while MS-SiT superforms the two other gDL networks. However, a per region box plot (Fig 2) of its performance relative to the MS-SIT shows this is largely driven by improvements to two large regions. Overall, MoNet and the MS-SIT differ significantly for 10 out of 32 regions, with MS-SIT outperforming MoNet for 6 of these. We also evaluated the performance of the MS-SiT model by providing it with more inductive biases, via transfer learning from a model first trained on the larger UKB dataset (achieving 0.94 dice for cortical parcellation), increasing slightly the final performance.

## 4. Discussion

The novel MS-SiT network presents an efficient and reliable framework, based purely on self-attention, for any biomedical surface learning task where data can be represented on sphericalised meshes. Unlike existing convolution-based methodologies translated to study general manifolds, MS-SiT does not compromise on filter expressivity, computational complexity, or transformation equivariance (Fawaz et al., 2021). Instead, with the use of local and shifted attention, the model is able to effectively reduce the computational cost of applying attention on larger sampling grids, relative to (Dahan et al., 2022), improving phenotyping performance, and performing competitively on cortical segmentation. Compared to convolution-based approach, the use of attention allows for the retrieval of attention maps, providing interpretable insights into the most attended cortical regions (Fig 5), and the methodology's robustness to transformations enables it to perform well on both registered and native space data, removing the need for spatial normalisation using image registration.

---

6. Run on a different train/test split, (Gopinath et al., 2020)

7. (Ha and Lyu, 2022)

8. (Monti et al., 2016)

## Acknowledgments

We would like to acknowledge funding from the EPSRC Centre for Doctoral Training in Smart Medical Imaging (EP/S022104/1).

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

## Appendix A. Methods

### A.1. Network architecture details

The details of the MS-SiT architecture are provided in Table 3.

| MS-SiT levels | Sequence length $|F_{6-l}|$ | Attention windows | Window size $w_l$ | Level architecture |
|---|---|---|---|---|
| $l = 1$ | 20480 | 320 | 64 | linear 96-d - LN
$[w_l$ 64, dim 96, head 3$] \times 2$
merging - LN |
| $l = 2$ | 5120 | 80 | 64 | linear 192-d - LN
$[w_l$ 64, dim 192, head 6$] \times 2$
merging - LN |
| $l = 3$ | 1280 | 20 | 64 | linear 384-d - LN
$[w_l$ 64, dim 384, head 12$] \times 6$
merging - LN |
| $l = 4$ | 320 | 1 | 320 | linear 768-d - LN
$[w_l$ 320, dim 768, head 24$] \times 2$
merging - LN |

Table 3: MS-SiT detailed architecture. The MS-SiT model adapts the Swin-T architecture (Liu et al., 2021) into a 4-level network with $\{2, 2, 6, 2\}$ local-MHSA blocks and $\{3, 6, 12, 24\}$ attention heads per level. As in (Liu et al., 2021), the initial embedding dimension is $D = 96$. Thus, the MS-SiT encoder has 27.5 M trainable parameters.

### A.2. Positional Embeddings

The MS-SiT model implements positional encodings in the form of 1D learnable weights, added to each token of the input sequence $X^0$. This follows the implementations in (Dosovitskiy et al., 2020; Dahan et al., 2022).

### A.3. Segmentation pipeline

The MS-SiT architecture can be turned into a U-shape network for segmentation for segmentation tasks Figure 3.

### A.4. Shifted-attention

An illustration of the shifted-attention mechanism is presented in Figure 4.

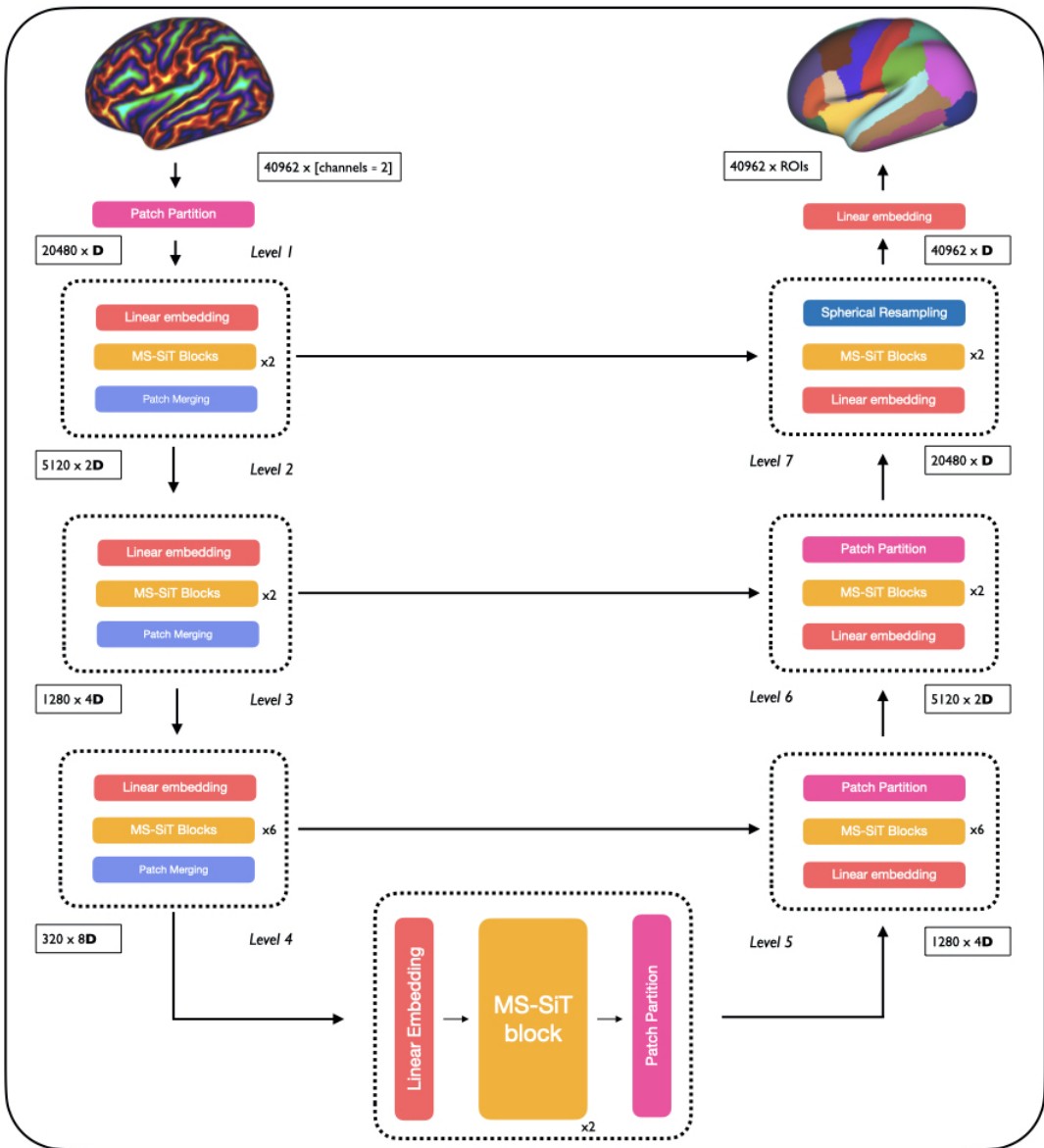

Figure 3: MS-SiT segmentation pipeline. Input data is resampled and partitioned as in Figure 1. The $l = \{1, 2, 3, 4\}$ levels of the segmentation pipeline are similar to the MS-SiT encoder levels (Figure 1). The *patch partition* layers reverse the patch merging procedure of the MS-SiT encoder, upsampling the spatial resolution of patches from $I_2 \rightarrow I_3 \rightarrow I_4 \rightarrow I_5$. Skip connections between levels are used. Finally, a *spherical resampling* layer resamples the final embeddings to an ico6 tessellation (40962 vertices), before the final segmentation prediction.

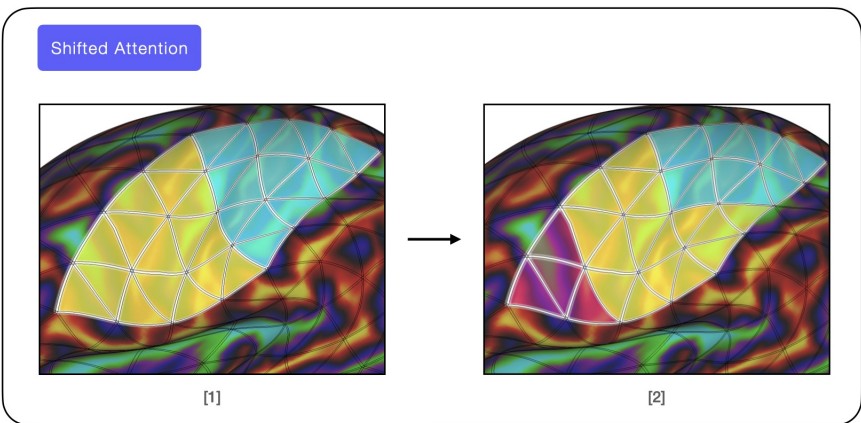

Figure 4: [1] W-MHSA applies self-attention within a local window, defined by a fixed regular icosahedral partitioning grid. Two local windows are show here, delimited by the yellow and blue colours here. [2] SW-MHSA shifts patches such that local attention is computed between patches originally in different local windows.

## Appendix B. Results

### B.1. Attention weights

Attention weights from the last layer of the MS-SiT can be extracted and visualised on the input surface, see Figure 5. They are compared to the attention weights extracted from a SiT model on the same GA prediction task.

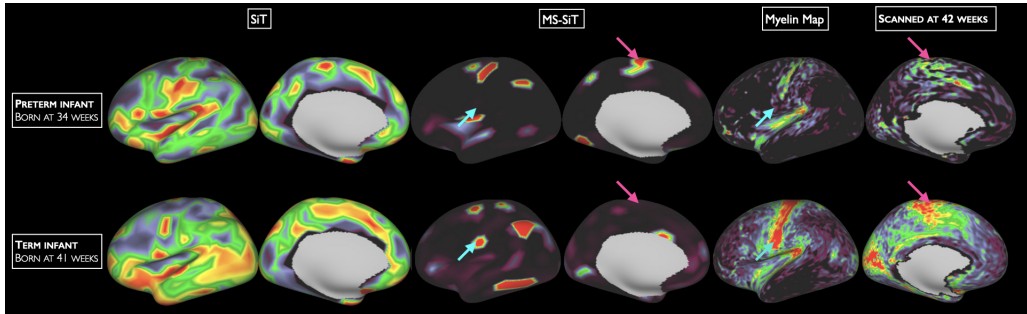

Figure 5: Comparison of normalised attention maps from the last layers of a SiT model (Fawaz et al., 2021) (applying global attention for all layers) and an MS-SiT model, both trained for GA-template prediction. MS-SiT maps display highly specific attention patterns, compared to the SiT counterparts, focusing on characteristic landmarks of cortical development such as the sensorimotor cortex with low myelination in preterm (pink arrows) and high myelination in term (blue arrows).

**B.2. Optimisation and hyperparameter search**

B.2.1. Hyperparameter search for optimal shifting factor

In table 4, we evaluate the impact of the shifting factor $w_s$ on the prediction performance.

| Shifted attention | Shift factor $w_s$ | MAE | MAE (augmentation) |
|---|---|---|---|
| ✗ | ✗ | $1.68 \pm 0.13$ | $1.24 \pm 0.04$ |
| ✓ | 1/2 | $1.55 \pm 0.03$ | $\mathbf{1.16 \pm 0.02}$ |
| ✓ | 1/4 | $\mathbf{1.54 \pm 0.08}$ | $1.20 \pm 0.03$ |
| ✓ | 1/16 | $1.56 \pm 0.09$ | $1.26 \pm 0.04$ |

Table 4: Hyper-parameter tuning of the shift factor $w_s$ in the **SW-MHSA** module. The use of $w_s \in \{\frac{1}{2}, \frac{1}{4}, \frac{1}{16}\}$ is compared to no shift for MS-SiT models trained to predict GA from template-aligned dHCP data. Models were trained for 25k iterations ($\sim$ 50% typical training runtime). We report MAE and std for the validation dataset, averaged over 3 runs are reported. Shifting the sequence of half the length of the attention windows, i.e. $w_s = \frac{1}{2}$, provides the best results overall and is used in all the following.

B.2.2. Optimisation and scheduling

The training strategy used for each task is summarised in Table 5. Extensive experiments showed that AdamW (Kingma and Ba, 2017) with linear warm-up and cosine decay scheduler was the best optimisation strategy overall (PyTorch Library - CosineAnnealingLR and GitHub - PyTorch Gradual Warmup). This follows standard practices (Gotmare et al., 2018) and training results from similar transformer models (Liu et al., 2021). Of note, SGD with momentum and small learning rate ($LR = 1e^{-5}$) also achieved good performances on the phenotype prediction tasks but could not converge on the cortical parcellation. Mean Square Error loss (MSE) was used to optimise models on the regression tasks and an unweighted combination of DiceLoss and CELoss (MONAI implementation MONAI - DiceCELoss) is used for optimisation of the segmentation task. We used a batch size of 16 for the phenotyping prediction experiments, and a batch size of 1 for the segmentation experiment (as it led to better results). In Figure 6, we compare the training and validation losses between our MS-SiT methodology and the SiT methodology.

| Model | Task | Optimiser | Warm-up it | Learning Rate | Training epochs |
|---|---|---|---|---|---|
| MS-SiT | PMA | AdamW/SGD | 1000 | $1e^{-5}$ | 1000 |
| MS-SiT | GA | AdamW/SGD | 1000 | $1e^{-5}$ | 1000 |
| MS-SiT | Cortical Parcellation | AdamW | 100 | $3e^{-4}$ | 200 |

Table 5: Training strategies for all tasks. Overall, AdamW with linear warp-up and cosine decay is selected as the default optimisation startegy.

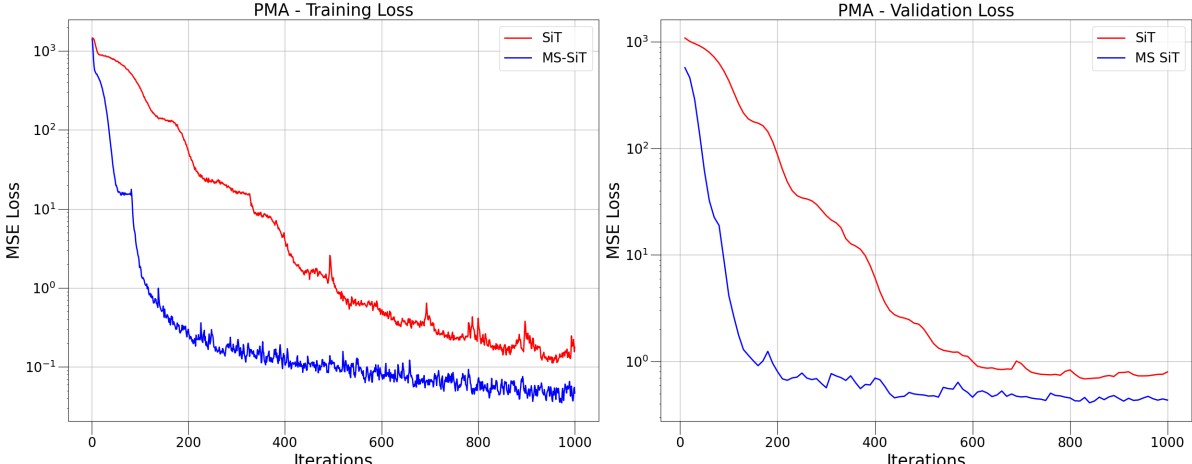

Figure 6:  Comparison of training and validation losses between MS-SiT and SiT-tiny (ico2) models trained on PMA. Plots seem to indicate a faster convergence of the MS-SiT model.

