# OpenReview forum: "The Multiscale Surface Vision Transformer"
_MIDL.io/2024/Conference — MIDL 2024 Oral_

### Official Review · Reviewer_ozm5 · 2024-02-28

**Confidence:** 3
**Preliminary Rating:** 5
**Final Rating:** 5

**Summary:**

The authors propose Multiscale SiT (MS-SiT), a model that draws inspiration from Swin Transformers and builds on the Surface Vision Transformers (SiT) framework from their previous research. The method adapts the shifted local-attention technique by shifting the sampling grid across the surface allowing model to preserve global interactions between distant regions on the input surface.

**Strengths:**

The paper is well written with clear motivation. It presents an innovative method that incorporates the latest developments in the field, demonstrating an advancement over previous techniques and surpassing current leading methods such as MoNet and SPHARM-Net. The authors bolster the integrity of their research by providing open access code and model weights. Additionally, the experiments conducted across both native and template spaces. Data augmentations are used to improve transformation invariance. The careful consideration given to crucial machine learning aspects, such as ensuring a balanced distribution across training and test set partitions, underscores the thoroughness of the research approach.

**Weaknesses:**

Limitations of their work is not discussed. Baseline comparison are limited and does not justify the need for sophisticated methods.

The lack of use of a widely used dataset, such as the HCP young adult, limits the study's comparability with existing literature. This is particularly crucial for phenotyping prediction.

Minor:
Missing some key information, i.e. input data used for analysis. For cortical parcellation task, the authors clearly state which input features are used. However, it is unclear what is used for phenotypic prediction. All these metrics: sulcal depth, curvature, cortical thickness and T1w/T2w myelination? If so why not in the parcellation task.

**Detailed Comments:**

N/A

**Justification Of Final Rating:**

The authors have impressively addressed my concerns and other comments, significantly improving the quality of the paper. Their attentive and thorough responses validate my initial assessment of their work.

**Justification Of The Preliminary Rating:**

I believe this is an excellent research paper, with its strengths significantly outweighing any weaknesses. While some clarifications might be necessary, I would be eager to see it presented at the conference.

**Questions To Address In The Rebuttal:**

Given the emphasis on "Multiscale" in the title and the inspiration drawn from Swin Transformers, could you provide a more in-depth explanation of the multiscale concept as it applies to your work? Specifically, how do hierarchical representations contribute to the objectives and findings of your paper?

Could you provide further explanation on the following points:

1. What motivated the choice of a U-net like architecture?
2. Can you elaborate on the computational footprint of your approach?
3. What are the time and memory requirements to train these models?

---

> ### Author Response · Authors · 2024-03-18
>
> We would like to thank the reviewer for their careful review of our submission and their many constructive comments.
>
> **Points to address:**
>
> *Multiscale concept*: The main motivation for introducing the ‘multiscale’ concept in the surface transformer architecture comes from the fact that understanding phenotypes requires a global understanding of the brain and therefore using deep learning architectures that can effectively connect distant areas of the brain although not adjacent. However, one of the main limitations that were pointed out in the SiT paper [Dahan et al. 2022, MIDL] is the cost of the self-attention operation that limits the resolution at which patches are sampled, and therefore the modelling of subtle variations across the cortex, specifically for dense prediction tasks. The multiscale framework proposed in the Swin transformer paper [Ze Liu et al 2021] (but also concurrent work such as [*Multiscale Vision Transformers*
> , H.Fan et 2021]) allows to model the data across scales – i.e. at the highest resolution the model can learn to model high-resolution features and textures, but only apply attention within regional local windows. By merging neighbouring patches of extracted features across layers, the model downsamples the resolution at which the patches are represented but increases the receptive field of the self-attention operation and eventually learns to attend across the whole brain. This is essential for modelling subtle cortical patterns and, at the same time, long-range dependencies between cortical regions.
>
> *U-Net architecture*: The use of a U-Net architecture was motivated by a previous paper that implemented vision transformers for 2D medical image segmentation  (Hu Cao et al 2021). In general UNet-style architectures are widely used for such tasks as they balance robust localisation of objects with sharp segmentation of boundaries. They achieve this by inserting high-frequency information back into the network through skip connections, applied between the encoding and decoding paths. We added this reference in the method section.
>
> *Computational footprint, Time and Memory requirements*: We thank the reviewer for the opportunity to provide essential information regarding the computational cost and performance of our method. The MS-SiT in its default parametrisation as in Table 3 has 27.5M trainable parameters. This is significantly larger than the SiT-tiny used in [Dahan et al 2022, MIDL] -  with 5M params but smaller than the SiT-base model and its 86M parameters [Dahan et al 2022, MIDL] - and seems reasonable compared to today’s standard with transformer models. On a standard 24G memory NVIDIA GPU, we can use a relatively large batch size of 28 (taking 22G of memory), which allows for fast convergence. We added a figure in Appendix B to compare the training and validation losses of an MS-SiT trained on PMA prediction against a SiT-tiny (ico2). The MS-SiT achieves convergence after 450 epochs (corresponding to 2.5 hours of training) compared to the SiT model (around 800 epochs). Additionally, as pointed in the method section, the MS-SiT local self-attention operation reduces the complexity from $\mathcal{O}(|F_{6-l}|^2)$ to $\mathcal{O}(w_{l}|F_{6-l}|)$ with $w_{l} << |F_{6-l}|$, where $|F_{6-l}|$ corresponds to the length of the sequence and $w_{l}$ the size of the attention window. Therefore, where the SiT is limited in the sequence length because of the quadratic complexity of the self-attention operation, here the MS-SiT can be trained on an input sequence of 20480 tokens, in a similar time to an SiT-tiny (ico2) (trained with 320 patches) and improves the performance.
>
> **Other comments**
>
> *Missing information*: We apologise for the lack of clarity around the input data used. For phenotype predictions, we used four cortical metrics (sulcal depth, curvature, cortical thickness and myelin maps). We added a sentence in the main text to address the confusion. For the parcellation task, we trained the network purely on shape-based cortical metrics – specifically sulcal depth (which captures coarse-scale patterns of cortical folding – predominantly primary folds) and mean curvature (secondary and tertiary folds), as the DK atlas [Desikan et al. 2006] parcels the brain into regions that align with the primary folding patterns of the brain. We clarify this point in the main text section 3.2.
>
> *HCP dataset*: The use of HCP data would be indeed essential and will be integrated in future work. We already have tried to experiment on cognitive score regression notably fluid intelligence prediction. Preliminary results seem to indicate that the MS-SiT can achieve a good performance using memory task contrast maps. We obtained a 0.42 Pearson correlation score (CV - 5 folds) for ~1000 HCP subjects. This can be compared with GNN methods trained on functional connectivity data derived from individual ICA cortical parcellation, achieving 0.33 Pearson correlation [Dahan et al 2021].

---

### Official Review · Reviewer_PLqJ · 2024-02-28

**Confidence:** 4
**Preliminary Rating:** 5
**Recommendation:** Best Paper Award
**Final Rating:** 5

**Summary:**

This paper presents a method to perform brain surface analysis using deep learning. The main contribution is a network architecture implementing graph NN on a cortical surfaces, based on transformers. The authors describe a multi scale transformer implementation that achieves very good results.

**Strengths:**

The paper is well written and the method sound.

The figures are of high quality and the results are good and convincing.

The multiscale nature of the model is achieved through patch merging, instead of feature reduction. It would be interesting to compare the two in subsequent work.

**Weaknesses:**

Not many weaknesses, this is a good paper.

Quite a few details are missing but lots is already crammed into the limited available space. Extra details are provided in supplementary materials which are very useful.

One big detail that is missing is how the positional encoding is performed. I assume some sort of spherical angular encoding, but this ought to be explained.

**Detailed Comments:**

No further comments.

**Justification Of Final Rating:**

Strong paper with an interesting method and good performance. The authors did a good job addressing the reviewers' comment. They have added details on the position encoding part, which is quite important.

**Justification Of The Preliminary Rating:**

The paper is very well written and the method is interesting. The authors did a good job in describing the method in the limited space available. The experimental design is sound and the figures are high quality.

**Questions To Address In The Rebuttal:**

Explain position encoding.

**Special Issue:**

Yes

---

> ### Author Response · Authors · 2024-03-18
>
> We would like to thank the reviewer for their careful review of our submission and their constructive comments. We corrected the typos that were present in our first submissions and added some missing details in the Appendix.
>
> **Points to address:**
>
> *Explain positional embedding*: As done in the original SiT paper [Dahan et al 2022, MIDL] positional embeddings (PE) are learnt. This is implemented through initialising the PE tokens at random, to be optimised during the training. The latest literature in vision transformers seems to indicate that fixed positional embedding (notably in the form of sine-cosine embeddings) is becoming the default setting. This could be tested in future work. We updated the method section to add the mention of trainable PE and added some additional details regarding PE in Appendix A.2. Also, we have never thought about using spherical angular encoding, as suggested by the reviewer. This is a great suggestion and will be tested in future work.

---

### Official Review · Reviewer_E7Z4 · 2024-02-28

**Confidence:** 4
**Preliminary Rating:** 5
**Recommendation:** Oral
**Final Rating:** 5

**Summary:**

This paper presents a multi scale surface vision transformer to process cortical surface data in neuroimaging. This model reduces the computational cost of fulll-resolution self-attention by combining local-mesh-window for within region and shifted-window self-attention for between region information sharing. The authors test their MS-SiT backbone on a phenotyping prediction task on dHCP data and on cortical segmentation on MindBoggle and UK Biobank data.

**Strengths:**

I think this is an excellent paper overall. Its main strengths are:
- Relevant topic: Geometric deep learning approaches are important for many areas of biomedical imaging.
- To the best of my knowledge this is a novel contribution to the field.
- The paper is well written.
- The methodology is well explained and motivated.
- The experiments are well designed and executed.
- The results are convincing.

**Weaknesses:**

I do not see a major weakness of the paper worth mentioning here.
There are only a few minor points that come to my mind. These mostly concern some details in the experiments and a bit of background on the task settings. I will list those below under detailed comments.

**Detailed Comments:**

- Experiments and Results part (p5)
   - Could you briefly mention the intuition for choosing the opimiser and scheduler (or provide a reference)?
   - Are the two losses weighted in any way? Why did you choose both of them?
- 3.2. Cortical parcellation on UKB & MindBoggle p.8:
   - It might be nice to briefly outline (the intuition behind) the cortical segmentation task: E.g. what kind of data / information goes into the model? This might make this part more accessible to a broader readership.

**Justification Of Final Rating:**

Thanks a lot for addressing and clarifying the minor points I had mentioned. I think this is a well-written paper with a great presentation of methods and results. The method itself could have a great impact on the neuroimaging field.

**Justification Of The Preliminary Rating:**

I think this paper is novel, relevant, well written and executed. I am convinced it will be of practical importance to a lot of people in the field of cortical analysis neuroimaging and in particular for those interested in geometric deep learning approaches.

**Questions To Address In The Rebuttal:**

See comments above.

**Special Issue:**

Yes

---

> ### Author Response · Authors · 2024-03-18
>
> We would like to thank the reviewer for their careful review of our submission and their constructive comments and feedback on the methodology's impact.
>
> **Points to address:**
>
> *Explain the choice of optimizer and the scheduler*: We ran extensive experiments comparing the use of SGD and AdamW optimisers, across tasks, and with a wide range of different hyper-parameters. We found that AdamW with a learning rate and cosine decay (eta_min = 1e-5) with a linear learning rate warm-up was the best performing overall. This is consistent with standard practices in vision transformer papers that use AdamW and cosine warm-up scheduling [A. Gotmare et al 2018, Z. Liu et al 2021]. Briefly, for the regression tasks we found that AdamW (with LR=1e-5 and 1000 warm-up iterations) and SGD (with a small learning rate, typically 1e-5) were performing similarly. However, for the segmentation task, we found that SGD was unstable such that convergence was difficult to achieve even with a learning rate scheduling scheme (e.g. LR step decay). Overall, we selected the AdamW with cosine warm-up as the default optimisation setting. We added more details regarding the optimisation of the models as well as the training hyper-parameters used for the different tasks in Appendix B 2.2.
>
> *Details on the segmentation loss*: For optimisation of the segmentation task, we used the Dice Cross Entropy loss implementation of Monai. While developing the model, we compared the results against the use of dice or cross-entropy only and as expected it seems that the Dice cross-entropy loss was performing better in terms of final Dice. We did not use any particular weighting between the losses (default MONAI parameter), but we believe that this can be an important hyperparameter to finetune depending on the segmentation task. Some additional information was also added in Appendix B 2.2.
>
> *Intuition behind the cortical segmentation task and details about the data*: The DK atlas [Desikan et al. 2006] parcels the brain into regions that align with the primary folding patterns of the brain. We therefore trained the network purely on shape-based cortical metrics – specifically sulcal depth (which captures coarse-scale patterns of cortical folding – predominantly primary folds) and mean curvature (secondary and tertiary folds). Both these metrics are output by default from the FreeSurfer surface processing pipeline. We clarify this point in the main text section 3.2.

---

### Meta-Review · Area_Chair_qEMx · 2024-04-04

**Recommendation:** Accept (Oral)
**Confidence:** 5

**Metareview:**

All reviewers found that this paper is novel and addresses an important topic that the MIDL community will be interested in. The paper is written well and demonstrates intensive results clearly.

---

### Decision · Program_Chairs · 2024-04-06

Accept (Oral)